# The Impact of CSI SEEE Carbon Neutral Index Launched on Order Aggressiveness

Zihuang Huang , Xiaoyu Zhang and Kaifeng Li *

School of Economics and Management, China University of Mining and Technology, Xuzhou 221116, China; tbh244@cumt.edu.cn (Z.H.); ts22070009a31@cumt.edu.cn (X.Z.)
* Correspondence: lkf@cumt.edu.cn

**Abstract:** In the context of carbon peaking and carbon neutrality goals, in order to clarify the investment direction for investors, China Securities Index Co., Ltd. (CSI) has collaborated with the Shanghai Environmental Energy Exchange to develop the CSI SEEE Carbon Neutral Index (CSCNI), which has also played a leading role in the subsequent preparation of the Green Finance Index. The launch of this index has sparked research interest among scholars in stimulating investor order aggressiveness. This study employs event study methodology to examine the impact of the CSCNI launch on order aggressiveness. The sample companies are categorized into two groups: deep low-carbon and high-carbon reduction, with a focus on studying buy and sale order aggressiveness. The results indicate that the launch of CSCNI has mobilized order aggressiveness but has led to a negative stock price effect as investors anticipate an increase in environmental costs for the sample companies. Furthermore, we reveal that the long-term growth potential of the deep low-carbon field is more promising compared to the high-carbon reduction sector, making stocks in the deep low-carbon field more attractive. The launch of CSCNI has shown contrasting effects on the buy and sale order aggressiveness of investors, with the impact of the index announcement being more significant on the sample companies. This research provides valuable insights for evaluating the impact of green finance indices and contributes to the understanding of internal mechanisms. It provides an important reference for financial regulators to evaluate the development of the current green index. At the same time, it expands the domestic research on order aggressiveness, which studies the action mechanism of the stock price effect of the green stock index from the perspective of order aggressiveness.

**Keywords:** CSI SEEE carbon neutral index; order aggressiveness; stock price effect; event study

## 1. Introduction

Since 2020, China's commitment to hit peak carbon emissions before 2030 and attain carbon neutrality before 2060 is an important national strategy. In 2021, the State Council of the People's Republic of China issued the Opinions of the CPC Central Committee and the State Council on Fully, Accurately, and Comprehensively Implementing the New Development Concepts and Doing a Good Job of Carbon Peaking and Carbon Neutrality, as well as the Action Plan for Carbon Peaking before 2030, and provinces have formulated carbon peaking implementation plans based on the actual situation in their provinces. Green and low-carbon development has become a firm choice for China, but opinions vary on the impact of green development on the capital market. Chan and Walter (2014) suggest that corporate green technology innovation will increase stock liquidity, while Li (2024) finds that corporate green transformation will have a negative impact on stock prices. Whether investors recognize the importance of green low-carbon development and apply this understanding to their investment strategies remains to be seen. The carbon neutral, ESG, and other green indices are important tools and carriers to implement carbon neutrality and promote sustainable investment. On 22 September 2021, China Securities

Index (CSI) made an announcement regarding the upcoming launch of the CSI SEEE Carbon Neutral Index (CSCNI, code: 931755) on 21 October 2021. For the capital market to further play a role in serving green transformation and upgrading the economy, CSI and Shanghai Environment and Energy Exchange (SEEE) have actively cooperated to develop the CSCNI.

Chinese scholars often integrate green indices with other variables in their research. For instance, Zhang et al. (2023) conducted a study analyzing the impact and pathways of China's green finance development on carbon emissions. Their research indicates that green finance plays a significant role in curbing carbon emissions, with the implementation of green finance policies reinforcing carbon reduction effects. Furthermore, they suggested that the influence of green finance on carbon emissions is influenced more by administrative and public environmental regulations rather than market-based environmental regulations. In another study, Zhao and Luo (2024) explored the correlation between climate uncertainty and the volatility of green indices. Their findings reveal that Chinese climate uncertainty and policy uncertainty are strong predictors of green index fluctuations.

In comparison to their Chinese counterparts, foreign scholars have conducted more extensive research in the field of green indices, resulting in a broader spectrum of conclusions. Cheung (2011) utilized U.S. stock data and observed that upon the announcement of the Dow Jones Sustainability World Index (a green index), the stocks included in it exhibited significant positive returns on the same day. On the contrary, Oberndorfer et al. (2013) found that in the German market, the launch of the same index triggered a negative reaction as companies considered the potential environmental compliance costs, leading to negative abnormal returns for related stocks. Additionally, Curran and Moran (2007) discovered that companies did not experience substantial rewards or penalties from being included or excluded from the FTSE Social Responsibility Index (another green index), with the market displaying a muted response to these events. The concept of ESG development has attracted much attention from investors, who identify companies with high social responsibility in their investment decisions, and enterprises with better ESG performance are more likely to be favored by investors (Renneboog et al. 2008). Investment institutions can obtain both financial utility and non-financial utility in line with personal and social values by investing in enterprises with good ESG performance (Bollen 2017). Given the insights from foreign research on green indices, the question arises: what impact has the launch of the CSI SEEE Carbon Neutral Index (CSCNI) had on the Chinese stock market?

To investigate the impact of the launch of the CSI SEEE Carbon Neutral Index (CSCNI) on the Chinese stock market, we can employ the concept of stock price effect, as defined by Oehler et al. (2017), which refers to the abnormal returns generated by stocks following a significant event. If component stocks of the CSCNI exhibit notable abnormal returns post-launch, it can be considered a stock price effect. A positive abnormal return signifies a positive stock price effect, while a negative abnormal return indicates a negative stock price effect. Previous studies have demonstrated the prevalence of stock price effects during pivotal events, particularly those with substantial industry influence like the introduction of green indices. For instance, Baulkaran (2019) conducted an empirical analysis and found that companies issuing green bonds experienced significantly positive cumulative abnormal returns, suggesting that issuing green bonds can enhance corporate value. Furthermore, Shaik (2021) identified a significant impact of the COVID-19 pandemic on the TASI stock market returns. To explore the stock price effect of the CSCNI launch, we conducted an event study to analyze the abnormal returns of sample companies following the index's introduction.

The Chinese stock market is known for being heavily influenced by policies and events, making unexpected occurrences crucial for investors. Following the release of the CSCNI, the People's Bank of China issued the Guiding Opinions on Further Strengthening Financial Support for Green and Low-carbon Development. We strongly believe that China's green financial product system will be further strengthened. In light of the country's carbon peaking and neutrality targets, this index has the potential to steer capital towards decar-

bonization efforts by shifting investor focus towards green and low-carbon initiatives. To better understand investor behavior, we introduce the concept of investor order aggressiveness (Biais et al. 1995), which provides more direct insight into investors' decision-making compared to traditional proxy variables like the Baidu Index and Google search index. By analyzing market order books, we are able to filter out noise from individual actions. Traditional proxy variables like the Baidu Index and Google search index contain a certain amount of noise, and there may be a long logical chain from investors' search, posting, and other behaviors to security-trading behaviors, and the relevant data are generally difficult to obtain through public channels. Even if investors search for relevant stocks, they may not participate in stock trading, which does not accurately reflect the attention of traders. Through panel regression modeling, we are able to validate the impact of investor order aggressiveness on market dynamics.

In the stock market under the efficient market hypothesis, every investor has the same information acquisition and mining ability and will not follow the trend of operation and irrational behavior, but this is inconsistent with the actual market situation. The majority of investors in China are retail investors who are limited by their personal professional level so that, even if they obtain the same information, they will cause information asymmetry due to cognitive differences, and because of the influence of subjective preferences, they will take some irrational investment behaviors, which is not conducive to the healthy operation of the stock market. Therefore, we should pay attention to the impact of green stock indices on investors so as to better guide the capital flow to green and low-carbon enterprises. We investigate the impact of the launch of the CSCNI on stock prices, offering valuable insights for policymakers assessing the development of green indices. In contrast to traditional methods using the Baidu Index and Google search index, we delve into the mechanism of the stock price effect following the introduction of the CSCNI from the perspective of order aggressiveness, contributing to the expansion of research on this topic in China. Our study reveals that while the release of the index heightened investors' order aggressiveness, it also resulted in a negative stock price effect. Traders are concerned that inclusion in the index may lead to increased environmental costs, consequently reducing profits. The statistically significant regression coefficients post-index launch indicate that investor order aggressiveness can partially account for the cumulative abnormal returns of the sampled companies.

The structure of the remaining text is outlined as follows: The second part entails analyzing the influence of the CSCNI release on component stocks through the lens of investor order aggressiveness and formulating research hypotheses. The third segment delves into data and methodologies, detailing the data sources, outlining the parameter settings for event study methodology, and elucidating the construction of variable order aggressiveness. The fourth section is further divided into three subsections, examining the presence of stock price effects and the validity of utilizing order aggressiveness to elucidate these effects, and showcasing the outcomes of robustness tests. The conclusion subsequently encapsulates the findings of this analysis.

## 2. Hypothesis

Amidst the context of carbon peaking and carbon neutrality goals, the CSCNI offers investors a clear investment trajectory, prompting active engagement from investors in trading component stocks. Kim et al. (2014) conducted a study on social responsibility disclosure and found that companies disclosing social responsibility information are less likely to experience future stock price collapses. Inclusion in the index signifies a reduced likelihood of environmental incidents for companies, thereby mitigating the risk of penalties for such occurrences, ultimately lowering environmental risk exposure. Conversely, while index selection may lead to increased environmental costs for companies, these costs can be offset over time through reduced financing expenses. Griffiths et al. (2000) found that order aggressiveness is driven by information content. Smales (2016) found a sharp increase in the number of orders submitted in the period following the policy announcement, and that

individual investors were more aggressive in their orders, focusing only on the likelihood of order execution. Being included in the index contains positive information about the long-term sustainable development of enterprises, which is conducive to increasing investor order aggressiveness.

**H1.** *The release of CSCNI will increase investor order aggressiveness.*

Scholars have engaged in ongoing discussions regarding the impact of inclusion in green indices on component companies. Cheung (2011) observed, using U.S. stock data, that upon the announcement of the Dow Jones Sustainability World Index (a green index variant), included stocks recorded significant positive returns. In contrast, Oberndorfer et al. (2013) noted a negative market response in the German market following a similar index unveiling, attributed to concerns over environmental costs, resulting in negative abnormal returns for relevant stocks. López et al. (2007) and Oberndorfer et al. (2013), in comparative analyses of European firms included in the Dow Jones Sustainability Index versus the Dow Jones Global Index, identified a negative short-term correlation between Dow Jones Sustainability Index inclusion and company performance. Cheung (2011) conducted a study utilizing the Dow Jones Sustainability World Index, analyzing the profit fluctuations of companies included or excluded from the index between 2002 and 2008. The findings revealed that following the announcement of inclusion or exclusion, short-term stock returns experienced significant increases or decreases, albeit transient in nature. The establishment of the CSCNI was a response to the carbon peaking and carbon neutrality objectives, with approximately one-third of its component stocks originating from High-carbon Reduction sectors. While index inclusion may entail a rise in companies' environmental costs, a study by Shaik (2021) amidst the global COVID-19 pandemic suggested a potential market downturn, potentially resulting in a negative reaction among sample stocks. Nevertheless, given the proliferation of national initiatives advocating for low-carbon development policies, investors may perceive sample stocks positively, potentially mitigating the negative stock price effects.

**H2.** *The release of CSCNI results in a negative stock price effect.*

Within the framework of the carbon neutrality strategy, industries traditionally associated with high-carbon emissions such as electricity, construction, industrial production, transportation, and agriculture are undergoing a transition towards low-carbon practices. This shift is imperative for fostering sustainable economic growth over the long term. The deep low-carbon sector encompasses enterprises engaged in renewable energy generation, nuclear power, energy management, water treatment, waste treatment, and related fields, positioning it as a burgeoning industry. LaBahn and Krapfel (2000) showed that by adopting clean production technology and renewable clean materials developed in green research, enterprises can achieve energy conservation and emission reduction and improve total factor productivity. Xiao et al. (2024) found that the potential of the low-carbon city pilot has a significant contribution to enterprise green technology innovation. As the backbone of enterprise innovation drive, low-carbon transformation has an important strategic position on a global scale. The development of a low-carbon economy is crucial for China to address energy resource constraints, ensure energy security, and combat environmental and climate challenges, presenting itself as an inevitable path for the nation. The long-term growth prospects of the deep low-carbon sector appear robust and promising.

**H3.** *Investors are more bullish on stocks in the deep low-carbon field compared to the high-carbon reduction field.*

In alignment with the objective of reaching a carbon dioxide emission peak before 2030 and harnessing the capital market's role in facilitating the economy's transition towards sustainability, CSI has introduced various green indices. Among these, the CSCNI stands

out as a pivotal tool and platform for the capital market to enact carbon-neutral strategies and foster sustainable investment practices. The process of index release encompasses both announcement and formal listing stages. Notably, foreign scholars have highlighted the presence of significant announcement premium effects in the stock market triggered by monetary policy announcements (Savor and Wilson 2013; Lucca and Moench 2015; Ai and Bansal 2018; Ai et al. 2022). The phenomenon of announcement premium is rooted in the unforeseen nature of monetary policy announcements, which introduce market uncertainty, necessitating increased risk premium compensation for investors. Similarly, announcements of index releases, compared to the subsequent listings, serve as unexpected market signals, exerting a more pronounced impact on the companies within the sample. Therefore, it can be inferred that the announcement of an index launch has a greater influence on sample companies compared to the actual listing of the index.

**H4.** *Compared to the launch of an index, the announcement of an index has a greater impact on the sample companies.*

### 3. Data and Methodologies
#### 3.1. Data

We acquired the constituent stocks of the CSCNI through CSI, a prominent financial market index provider established through the joint efforts of the Shanghai Stock Exchange and the Shenzhen Stock Exchange in August 2005. CSI has garnered significant domestic and global influence, managing a portfolio of over 7000 indices by the conclusion of 2023. These indices span various asset classes, including stocks, bonds, commodities, and funds across more than 20 major countries and regions worldwide, with a particular focus on the Shanghai, Shenzhen, and Hong Kong markets. On 22 September 2021, CSI announced the forthcoming launch of the CSI SEEE Carbon Neutral Index on 21 October 2021. The CSCNI comprises 100 securities selected from listed companies operating in the deep low-carbon sector, such as clean energy and energy storage, as well as the high-carbon reduction sector encompassing industries like thermal power and steel. These companies exhibit substantial market capitalization and demonstrate significant potential for carbon emission reduction, positioning them as key index constituents. The primary objective of the index is to gauge the performance of securities that make notable contributions to carbon neutrality. The introduction of the CSCNI marks a significant milestone in the seamless integration of the carbon market and the capital market, facilitating the redirection of social investments towards enterprises transitioning to a low-carbon economy. Consequently, we have identified the launch of the CSCNI as the focal point of our research event. Table 1 outlines the key characteristics of the CSCNI index.

**Table 1.** Index characteristics On 5 April 2023 on the CSI website.

|  | Market Value (CNY 100M) |
| --- | --- |
| Index Free Float Market Cap | 42,679.43 |
| Constituent Largest Free Float Market Cap | 5686.17 |
| Constituent Smallest Free Float Market Cap | 17.26 |
| Constituent Mean Free Float Market Cap | 426.79 |
| Constituent Median Free Float Market Cap | 229 |

We have chosen the index constituent stocks as our sample companies, specifically focusing on those that remained as index constituents as of 5 April 2023, while excluding companies with incomplete or missing data. In total, our sample comprises 71 companies, which have been categorized into two distinct groups based on the index methodology: deep low-carbon and high-carbon reduction. This classification allows for a clear delineation between companies operating in the deep low-carbon sector, characterized by

sustainable practices and low-carbon emissions, and those in the high-carbon reduction sector, which are actively engaged in reducing their carbon footprint.

We sourced the data for our sample companies from the Wind database, while the order book data were obtained from Level 2 data. Both the Shanghai Stock Exchange (SSE) and Shenzhen Stock Exchange (SZSE) offer Level 2 data to users through stock information service providers, which encompasses detailed order book information such as transaction-by-transaction specifics, top ten market depth, total order quantity, and weighted prices. The exchanges have continuously enhanced the comprehensiveness of the order book data provided to users, incorporating more frequent updates over time. Our primary focus lies on utilizing transaction-by-transaction details extracted from the order book data for our analysis.

Each trading day in mainland China is segmented into distinct periods: the opening call auction period (9:15 to 9:30), the continuous trading period (9:30 to 11:30, 13:00 to 14:57), and the closing call auction period (14:57 to 15:00). To ensure clarity and accuracy in our analysis, this study exclusively focuses on data gathered during the morning session from 9:30 to 11:30 and the afternoon session from 13:00 to 14:57 on each trading day. This selective approach helps us avoid any potential confusion arising from non-official trading data and ensures that our analysis is based on the most relevant and reliable information available during these specific trading periods.

*3.2. Event Study*

We employ an event study methodology to evaluate the abnormal returns in the stock value of a selected company associated with the events surrounding the launch of the CSCNI. In this study, we identify two key events: Event 1, corresponding to the announcement made on 22 September 2021, and Event 2, linked to the index's launch on 21 October 2021. The event dates considered for the event study analysis are 22 September 2021 and 21 October 2021. For the estimation of abnormal returns, we utilize an estimation period of 30 trading days, commencing 30 trading days prior to Event 1 and concluding on the actual Event 1 day, as outlined in Table 2. Additionally, we extend the estimation window for robustness testing by including a broader timeframe. Specifically, we incorporate a one-week window around the event day (0), encompassing 5 trading days before (−5) and after (5), resulting in a total of 11 trading days, following Xi and Jing (2021). Considering the anticipated, immediate, and delayed effects of the event, as highlighted by Zou et al. (2020), we conduct a comprehensive analysis to evaluate the short-term impact of the event on the sample company's returns. This approach allows us to capture the nuanced effects of the events on stock returns over different time horizons, providing a more thorough assessment of the event's influence on the company's financial performance.

**Table 2.** Specific settings for event window.

|  | Estimation Window | Event Day | Event Window |
|---|---|---|---|
| Event 1 | 2 August 2021–12 September 2021 | 22 September 2021 | 13 September 2021–28 September 2021 |
| Event 2 | 2 August 2021–12 September 2021 | 21 October 2021 | 14 October 2021–27 October 2021 |

To compute abnormal returns, we employ an event study analysis by comparing the actual returns during the event window with the mean returns observed during the estimation period of 30 trading days. In order to enhance the reliability of our analysis, we conducted additional tests using the CSI 300 Index as a benchmark for normal returns. This comparative approach allows us to assess the abnormal returns more accurately and provides a robust framework for evaluating the significance of the events on the company's financial performance.

$$AR_{i,t} = R_{i,t}^* - E(R_i) \qquad (1)$$

where $AR_{i,t}$ is the abnormal returns for stock i on time t, $R_{i,t}^*$ is the realized stock return for stock i on time t, and $E(R_i)$ is the average stock return.

$$CAR_{i,t} = \sum_{t=1}^{T} AR_{i,t} \tag{2}$$

where $CAR_{i,t}$ is the cumulative abnormal returns for stock i from time t to T (Dawar Gaurav and Parkash 2023).

Finally, a hypothesis test is constructed to identify whether this event had a significant impact on the stock returns of index constituents. We assume that the abnormal returns are distributed normally with a mean of zero. Hence, the null hypothesis is modified as follows:

$$t_{AR} = \frac{\frac{mean(AR_{i,t})}{S(AR_{i,t})}}{\sqrt{m}} \tag{3}$$

$$t_{CAR} = \frac{\frac{mean(CAR_{i,t})}{S(CAR_{i,t})}}{\sqrt{m}} \tag{4}$$

where m is the number of sample stocks; $S(\cdot)$ is the standard deviation.

### 3.3. Investor Order Aggressiveness

Based on the classification proposed by Biais et al. (1995), order aggressiveness is determined by analyzing the order price in relation to the market conditions. Specifically, orders are categorized based on whether the price is below or on the opposite side of the market, as well as whether the price falls within or above the same side of the market. The classification of order aggressiveness is as follows:

1.   Most aggressive sale orders: these orders aim for immediate execution by offering a price below Ask1 (the lowest price in all unfilled sale orders) and are assigned a value of 1.
2.   Aggressive sale orders: orders with prices between Ask1 and Ask5 (the fifth lowest price in all unfilled sale orders) are considered aggressive and are queued with a value of 2.
3.   Non-aggressive sale orders: orders priced above Ask5, while still queued, are not immediately visible to traders and are assigned a value of 3.
4.   Least aggressive sale orders: orders with prices above Ask10 (the tenth lowest price in all unfilled sale orders) are deemed the least aggressive and are assigned a value of 4.

This classification system, as outlined by Bian et al. (2018), utilizes levels from 1 to 4 to represent the degree of order aggressiveness, with Level 1 indicating the most aggressive orders and Level 4 indicating the least aggressive orders. A higher level corresponds to a lower degree of aggressiveness in the order placement, providing a framework for understanding the relative aggressiveness of different types of sale orders in the market.

We utilize Level 2 data to calculate the order aggressiveness in 10 min intervals. The calculation method is as follows:

$$OA_{i,n,t} = \frac{\sum_{k=1}^{4}(k_{i,n,t} \times Volume_{i,n,t}^{k})}{\sum_{k=1}^{4} Volume_{i,n,t}^{k}} \tag{5}$$

where $OA_{i,n,t}$ is investor order aggressiveness in time t, and the smaller $OA_{i,n,t}$, the more aggressive the order is; $Volume_{i,n,t}^{k}$ denotes the order quantity in time t; $n = b, s$ represents the buy and the sale; $k_{i,n,t}$ is the order aggressiveness level; and $k_{i,n,t} = 1, 2, 3, 4$.

*3.4. Regression Model*

To validate the rationale behind explaining stock price effects from the perspective of investor order aggressiveness, according to Aditya et al. (2000), we establish the following regression model:

$$CAR_{i,t} = \beta_0 + \beta_1 OA_{i,n,t} + lnMV_{i,t} + \varepsilon_{i,t} \tag{6}$$

We use Equation (6) to measure the impact of investor order aggressiveness on the cumulative abnormal returns during the event window. To maintain data frequency consistency, $AR_{i,t}$ is calculated at 10 min intervals, and the cumulative abnormal returns $CAR_{i,t}$ are calculated based on Equation (2); $OA_{i,n,t}$ is the order aggressiveness at time t. $lnMV_{i,t}$ is the logarithm of the market value.

*3.5. Robustness Testing*

3.5.1. Recalculate the Abnormal Returns

Given the potential influence of different models on the test outcomes, we opt to utilize the average returns of the Shanghai and Shenzhen 300 Index during the estimation window for computing abnormal returns. By doing so, we aim to reassess the significance of the abnormal returns and cumulative abnormal returns in light of this revised approach. The CSI 300 Index comprises the 300 largest and most liquid A-share stocks, serving as a representation of the overall performance of the China A-share market.

By incorporating the average returns of the Shanghai and Shenzhen 300 Index within the estimation window, we can enhance the accuracy and reliability of our abnormal return calculations. This adjustment allows us to better evaluate the impact of the events under consideration on the sample company's stock returns by comparing them against the performance of the broader market represented by the CSI 300 Index. The other parameter settings remain consistent with those previously mentioned, ensuring a systematic and comprehensive analysis of the abnormal returns and their significance in the context of the events surrounding the launch of the CSCNI.

3.5.2. Extend the Estimated Window Length

The length of the estimation window plays a crucial role in determining the abnormal returns and their subsequent impact on the cumulative abnormal returns. In order to assess the sensitivity of our results to the estimation window, we have decided to extend the event estimation window to 120 trading days preceding the event window. This adjustment allows for a more comprehensive analysis of the abnormal returns by incorporating a longer time frame for estimating the normal returns against which the event effects are evaluated.

By extending the estimation window to 120 trading days before the event window, we aim to capture a broader range of market dynamics and trends that may influence the stock returns of the sample company. This expanded window provides a more robust basis for calculating abnormal returns and enhances the reliability of our analysis of the cumulative abnormal returns. It is important to note that all other test settings will remain consistent with those outlined in the main text, ensuring a systematic and rigorous evaluation of the abnormal returns and their significance in the context of the events surrounding the launch of the CSCNI.

3.5.3. Shorten the Time Interval

Given the potential variability in investors' order aggressiveness across different time intervals, we have decided to adjust the calculation interval from 10 min to 5 min. This modification is aimed at mitigating the influence of time intervals on the experimental results and enhancing the credibility of our findings. By reducing the calculation interval to 5 min, we can capture more granular data on order aggressiveness, allowing for a more detailed analysis of investors' trading behavior and its impact on market dynamics.

This refinement in the calculation interval will enable us to better understand the fluctuations in order aggressiveness over shorter time frames and identify any patterns or trends that may not be apparent with a longer interval. By incorporating more frequent data points, we can gain deeper insights into the dynamics of order flow and its implications for market efficiency and price discovery. This adjustment will help to improve the accuracy and robustness of our analysis, ensuring that our results are more representative of investors' behavior across different time intervals.

## 4. Results

### 4.1. Existence of Stock Price Effect

We conducted tests on the stock price effects of both the announcement (Event 1) and the launch (Event 2) of the CSCNI. To clearly present the test results and analyze the trend in cumulative abnormal returns, we plotted $AR_{i,t}$ and $CAR_{i,t}$ in Figure 1 for the event window. Regardless of whether it is Event 1 or Event 2, we observed a consistent downward trend in cumulative abnormal returns after the event day. This trend indicates that the sample companies as a whole experienced negative abnormal returns following the announcement and launch of the CSCNI, suggesting that the introduction of the index had a negative impact on stock prices.

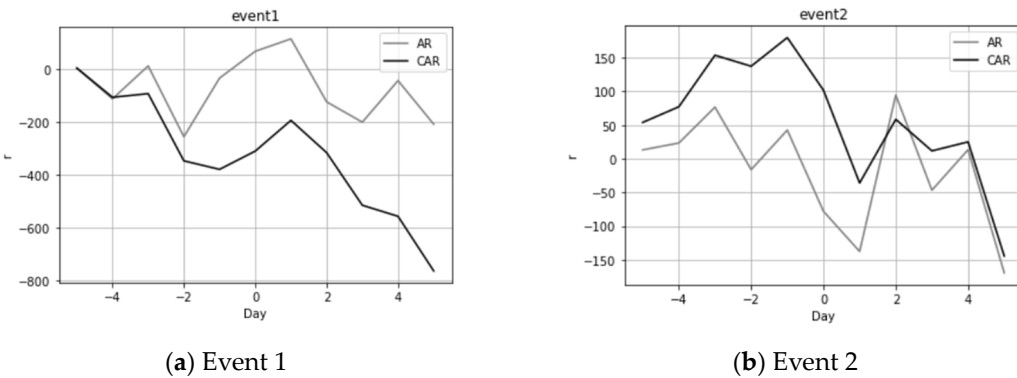

(**a**) Event 1         (**b**) Event 2

**Figure 1.** During the event window $AR_{i,t}$ and $CAR_{i,t}$ trend.

By plotting the abnormal returns and cumulative abnormal returns in Figure 1, we can visually illustrate the stock price effects of the events and examine the pattern of abnormal returns over the event window. The downward trajectory of cumulative abnormal returns highlights the overall negative impact of the CSCNI launch on the sample companies, underscoring the significance of the index launch in influencing stock price movements. This visualization provides a clear representation of the trend in abnormal returns and offers valuable insights into the market reaction to the events surrounding the CSCNI announcement and launch.

Table 3 presents the results of the T-test for $AR_{i,t}$ and $CAR_{i,t}$. Within the event window, $CAR_{i,t}$ for Event 1 is found to be significantly different from zero, indicating a notable impact on the sample companies' returns. In contrast, for Event 2, only one trading day's $CAR_{i,t}$ is significantly different from zero, suggesting a relatively weaker effect compared to Event 1. These results suggest that Event 1 had a more pronounced impact on the sample companies' returns compared to Event 2.

In order to further investigate whether the impact of the CSCNI release event on sample companies is only short-term and to verify if the CSCNI release leads to negative stock price effects, we analyzed abnormal returns and cumulative abnormal returns one month after the index release. In Figure 2, $AR_{i,t}$ for both Events 1 and 2 fluctuates around zero, while $CAR_{i,t}$ exhibits a distinct downward trend. Following Event 1, $CAR_{i,t}$ ceases decreasing from the +10th trading day onwards, displaying a gradual upward trend. In contrast, after Event 2, the downward trend diminishes from the +17th trading day onwards. These observations suggest that both events have an impact on the returns of

sample companies, with Event 2 demonstrating a longer duration of negative stock price effects compared to Event 1.

**Table 3.** $AR_{i,t}$ and $CAR_{i,t}$ T-test results.

| Day | Event 1 | | Event 2 | |
|---|---|---|---|---|
| | $AR_{i,t}$ (%) | $CAR_{i,t}$(%) | $AR_{i,t}$ (%) | $CAR_{i,t}$(%) |
| −5 | −0.31 | −0.31 | 0.34 | 0.34 |
| −4 | −1.32 *** | −1.64 *** | 0.32 | 0.67 |
| −3 | −0.01 | −1.65 *** | 0.62 | 1.29 |
| −2 | −4.01 *** | −5.66 *** | −0.33 | 0.96 |
| −1 | −0.24 | −5.90 *** | 0.75 | 1.70 |
| 0 | 2.10 *** | −3.80 *** | −1.39 *** | 0.31 |
| +1 | 0.04 | −3.76 *** | −1.74 *** | −1.43 |
| +2 | −1.84 *** | −5.61 *** | 1.33 *** | −0.10 |
| +3 | −3.5 *** | −9.14 *** | −1.18 *** | −1.28 |
| +4 | −0.38 | −9.52 *** | 0.32 | −0.96 |
| +5 | −1.86 *** | −11.38 *** | −2.29 *** | −3.25 *** |

Superscripts indicate statistical significance at the 1% (***) level.

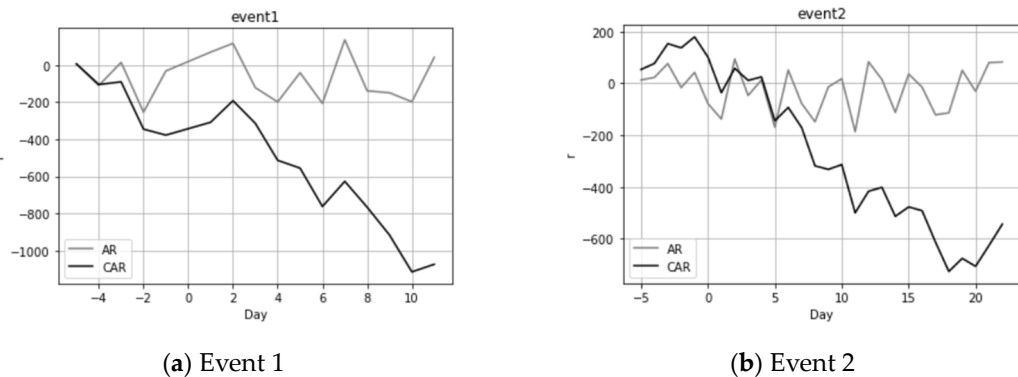

(**a**) Event 1                    (**b**) Event 2

**Figure 2.** Within one month after event $AR_{i,t}$ and $CAR_{i,t}$ trend.

However, over time, $CAR_{i,t}$ begins to reverse and rise, indicating that the impact of the index announcement and its formal release on stock prices is transient and manageable. The stabilization and eventual improvement in cumulative abnormal returns suggest that the initial negative effects of the CSCNI release event on stock prices are not enduring and can potentially be mitigated. These findings underscore the importance of monitoring the post-event dynamics of stock price effects to assess the lasting implications of market events and the ability of companies to navigate through transient fluctuations in returns.

In light of the CSCNI's objective to enhance investor focus on the green economy, we classified sample companies into deep low-carbon and high-carbon reduction sectors according to the index methodology to examine the effects of Events 1 and 2 on distinct company categories. As illustrated in Figure 3, abnormal returns for companies in the deep low-carbon and high-carbon reduction sectors hover around a mean of zero, indicating no significant deviations from expected returns. Meanwhile, $CAR_{i,t}$ displays a downward trajectory for both categories, suggesting a general decline in returns following the events. The observed patterns in abnormal returns and cumulative abnormal returns for companies in the deep low-carbon and high-carbon reduction sectors underscore the overarching negative impact of Events 1 and 2 on the sample companies, irrespective of their carbon reduction classification. The consistent downward trend in cumulative abnormal returns highlights a broad-based decline in returns across both categories, reflecting the market's response to the CSCNI release events. These findings provide valuable insights into the uniformity of stock price effects on companies categorized based on their carbon reduction efforts, indicating a collective market reaction to the introduction of the index and its implications for companies in the green economy space.

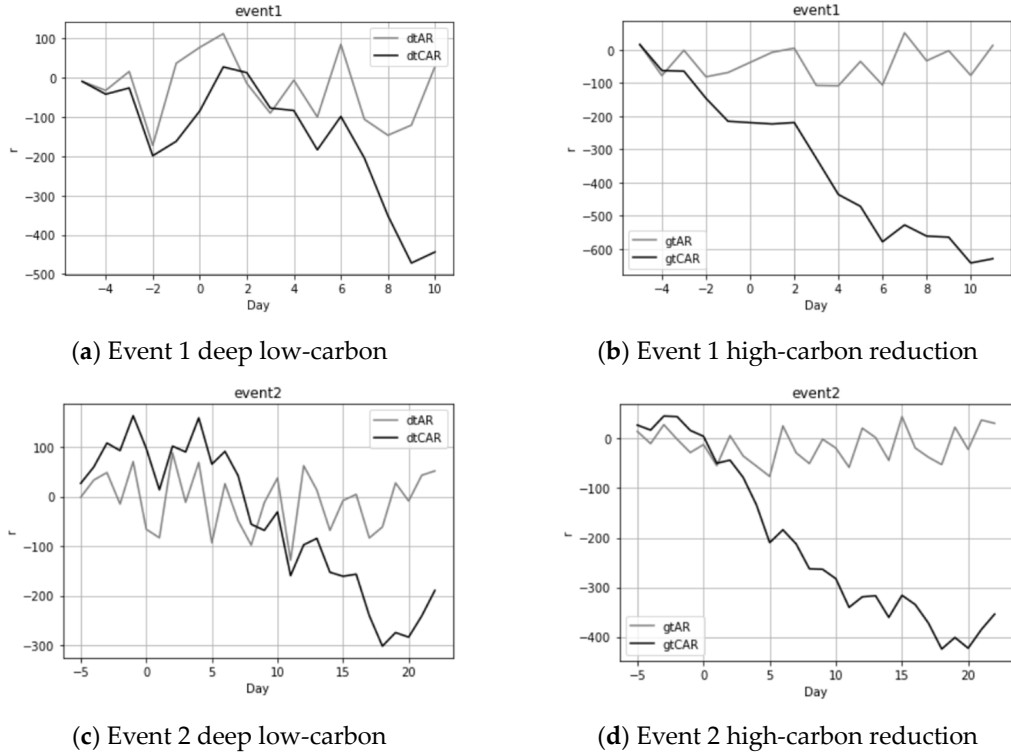

**Figure 3.** Thirty trading days after the event deep low-carbon and high carbon emission reduction sample $AR_{i,t}$ and $CAR_{i,t}$ trends.

In Figure 3a, $CAR_{i,t}$ of the deep low-carbon sector exhibits a slight increase one trading day after the event, followed by a minor decline from the +second to the +fifth trading day, a sharp drop on the +sixth trading day, and a subsequent rise from the +ninth trading day. Conversely, Figure 3b shows that $CAR_{i,t}$ of the high-carbon reduction sector started decreasing immediately after the event and continued to decline until the +ninth trading day without displaying an upward trend. These distinct patterns suggest that Event 1 has heightened investor interest in the deep low-carbon sector.

Similar trends are observed in Figure 3c,d, indicating that both Event 1 and Event 2 have stimulated investor attention and discussions in the low-carbon sector, leading to fluctuations in $CAR_{i,t}$ of the deep low-carbon sector. In contrast, opinions regarding the high-carbon reduction sector appear to remain relatively stable, resulting in a consistent decline in $CAR_{i,t}$ of the high-carbon reduction sector over the observed period.

The contrasting behaviors of the deep low-carbon and high-carbon reduction sectors post-events suggest that the market response to the CSCNI release events has triggered increased interest and volatility in the low-carbon sector, while sentiments towards the high-carbon reduction sector have remained more subdued. These findings underscore the differential impact of the events on companies within distinct carbon reduction categories, highlighting the varying levels of investor attention and sentiment towards different segments of the green economy.

*4.2. Regression Results*

4.2.1. Regression of Constituent Stocks as a Whole

To capture the shifts in investor sentiment and enthusiasm surrounding the index announcement, Table 4 presents a statistical summary of investor behavior around the event date. The key variables included in the analysis are as follows: $CAR_{i,t}$, representing cumulative abnormal returns; $OA_{I,t}$, indicating investor order aggressiveness; $OA_{i,b,t}$, denoting the buy-weighted average investor order aggressiveness; $OA_{i,s,t}$, representing the sale-weighted average investor order aggressiveness; and $lnMV_{i,t}$, reflecting the natural logarithm of the sample company's market value.

**Table 4.** Summary statistics.

| Title 1 | | Mean | Min. | Max. |
|---|---|---|---|---|
| Panel A: Event 1 | | | | |
| Pre-Announcement | $CAR_{i,t}$ | −0.41 | −11.29 | 10.05 |
| | $OA_{i,t}$ | 1.79 | 1.00 | 4.00 |
| | $OA_{i,b,t}$ | 1.78 | 1.00 | 4.00 |
| | $OA_{i,s,t}$ | 1.80 | 1.00 | 4.00 |
| | $lnMV_{i,t}$ | 15.67 | 13.28 | 18.65 |
| Post-Announcement | $CAR_{i,t}$ | −0.32 | −11.16 | 12.02 |
| | $OA_{i,t}$ | 1.74 | 1.00 | 4.00 |
| | $OA_{i,b,t}$ | 1.73 | 1.00 | 4.00 |
| | $OA_{i,s,t}$ | 1.74 | 1.00 | 4.00 |
| | $lnMV_{i,t}$ | 15.66 | 13.16 | 18.61 |
| Panel B: Event 2 | | | | |
| Pre-Announcement | $CAR_{i,t}$ | 0.58 | −6.97 | 11.31 |
| | $OA_{i,t}$ | 1.72 | 1.00 | 4.00 |
| | $OA_{i,b,t}$ | 1.73 | 1.00 | 4.00 |
| | $OA_{i,s,t}$ | 1.71 | 1.00 | 4.00 |
| | $lnMV_{i,t}$ | 15.61 | 13.14 | 18.77 |
| Post-Announcement | $CAR_{i,t}$ | −0.12 | −10.33 | 11.40 |
| | $OA_{i,t}$ | 1.72 | 1.00 | 4.00 |
| | $OA_{i,b,t}$ | 1.72 | 1.00 | 4.00 |
| | $OA_{i,s,t}$ | 1.73 | 1.00 | 4.00 |
| | $lnMV_{i,t}$ | 15.62 | 13.04 | 18.81 |

Based on the descriptive statistics in Table 4, the mean cumulative abnormal returns $OA_{i,t}$ before Event 1 stand at −0.41%, with a maximum of 10.05% and a minimum of −11.29%. Prior to and following Event 1, both the buy-side $OA_{i,b,t}$ and sale-side $OA_{i,s,t}$ show a decrease. In contrast, before and after Event 2, $OA_{i,t}$, $OA_{i,b,t}$, and $OA_{i,s,t}$ remain relatively stable. These findings suggest that the index announcement triggers a change in investors' order aggressiveness and prompts a forecast effect, leading to fluctuations in investor behavior. However, when the index is officially launched, there are no significant variations in investor order aggressiveness, indicating a stabilization in investor sentiment and behavior post-index release. The observed patterns in investor order aggressiveness before and after the index announcement highlight the dynamic nature of investor reactions to market events, with shifts in enthusiasm and behavior preceding the formal launch of the index. The relative stability in investor order aggressiveness following the index release underscores the establishment of a new equilibrium in investor sentiment and trading activity in response to the market developments.

The regression results of Equation (6) presented in Table 5 shed light on the relationship between investor order aggressiveness and abnormal returns following the release of the CSCNI. The significant coefficient values of $OA_{i,t}$ at the 1% level post-Events 1 and 2 suggest that the introduction of the index has a discernible impact on mobilizing investor order aggressiveness. Inclusion in the index is viewed as advantageous for companies' long-term sustainable development, attracting increased investor participation in trading activities.

Prior to Event 2, the persistence of a significant coefficient value for $OA_{i,t}$ can be attributed to the short interval of one month between Events 1 and 2. As a result, the window period before Event 2 still encapsulates information from Event 1, leading to a notable positive effect of $OA_{i,t}$ on $CAR_{i,t}$ within this timeframe. Following Event 2, the coefficient of $OA_{i,t}$ shifts to a significant negative value, indicating that lower levels of investor order aggressiveness correspond to reduced cumulative abnormal returns. This negative relationship suggests that investors anticipate an increase in the environmental costs of component companies, which could potentially erode profits and dampen returns. The observed dynamics between investor order aggressiveness and abnormal returns post-

Event 2 underscore the evolving perceptions and expectations of investors regarding the environmental implications and financial outcomes for companies included in the index.

**Table 5.** Regression results.

| | Event 1 | | Event 2 | |
|---|---|---|---|---|
| | Pre-Announcement | Post-Announcement | Pre-Release | Post-Release |
| $OA_{i,t}$ | −0.15 | 0.45 *** | 0.42 *** | −0.23 *** |
| | (−1.46) | (4.21) | (6.01) | (−2.87) |
| $lnMV_{i,t}$ | 47.41 *** | 17.19 *** | 47.22 *** | 41.73 *** |
| | (55.49) | (25.33) | (60.50) | (65.37) |
| Obs. | 8804 | 10625 | 8804 | 10650 |
| Adj.$R^2$ | 0.2586 | 0.1041 | 0.2520 | 0.2924 |

Superscripts indicate statistical significance at the 1% (***) level, and t-statistics are stated in parentheses.

### 4.2.2. Regression of Constituent Stocks Divided into Deep Low-Carbon and High-Carbon Reduction

We present the regression results of Equation (6) for the deep low-carbon sector and the high-carbon emission reduction sector in Table 6, following the segmentation of sample stocks. Post-Event 1, a notable shift is observed in the regression coefficient of $OA_{i,t}$ for the deep low-carbon sector, transitioning significantly from negative to positive. In contrast, the significance of the regression coefficient weakens for the high-carbon emission reduction sector. Similarly, after Event 2, the regression coefficient of $OA_{i,t}$ for the deep low-carbon sector undergoes a significant change from positive to negative, while the significance of the regression coefficient for the high-carbon emission reduction sector diminishes as well. These results suggest that Events 1 and 2 have heightened investor focus on the deep low-carbon sector relative to the high-carbon emission reduction sector.

**Table 6.** Regression results for deep low-carbon and high-carbon reduction.

| Panel A: Event 1 | | | | |
|---|---|---|---|---|
| | Deep low-carbon | | High-carbon reduction | |
| | Pre-Announcement | Post-Announcement | Pre-Release | Post-Release |
| $OA_{i,t}$ | −0.44 *** | 0.54 *** | 0.40 ** | 0.26 * |
| | (−3.82) | (3.75) | (2.27) | (1.80) |
| $lnMV_{i,t}$ | 48.44 *** | 22.54 *** | 42.91 *** | 25.26 *** |
| | (46.46) | (23.31) | (29.13) | (22.84) |
| Obs. | 5332 | 6450 | 3472 | 4175 |
| Adj.$R^2$ | 0.27 | 0.09 | 0.25 | 0.17 |
| Panel B: Event 2 | | | | |
| | Deep low-carbon | | High-carbon reduction | |
| | Pre-Announcement | Post-Announcement | Pre-Release | Post-Release |
| $OA_{i,t}$ | 0.38 *** | −0.34 *** | 0.45 *** | −0.06 |
| | (4.50) | (−3.75) | (3.72) | (−0.42) |
| $lnMV_{i,t}$ | 45.98 *** | 43.77 *** | 44.54 *** | 43.60 *** |
| | (48.57) | (48.09) | (31.64) | (41.68) |
| Obs. | 5332 | 6450 | 3472 | 4200 |
| Adj.$R^2$ | 0.30 | 0.25 | 0.19 | 0.35 |

Superscripts indicate statistical significance at the 1% (***), 5% (**), and 10% (*) levels, and t-statistics are stated in parentheses.

Following both events, the absolute value and significance level of the coefficient of $OA_{i,t}$ in the deep low-carbon sector surpass those in the high-carbon emission reduction sector. This disparity underscores the increased attention and order aggressiveness exhibited by investors towards companies in the deep low-carbon sector compared to those in

the high-carbon emission reduction sector following the CSCNI release. The release of the CSCNI has positioned the low-carbon economy as a strategic imperative for China to address climate and environmental challenges. The observed surge in investor confidence and order aggressiveness towards the deep low-carbon sector post-index release signifies a growing recognition of the sector's potential and sustainability. This heightened investor interest in low-carbon stocks underscores the transformative impact of the index on investor perceptions and market dynamics, driving increased attention and investment in companies aligned with low-carbon initiatives.

4.2.3. Regression of Investor Order Aggressiveness Divided into Sale and Buy

This part presents the results of studying the impact of the release of the CSCNI on investor order aggressiveness, further segmented into buy and sale directions, as shown in Table 7. The analysis reveals that $OA_{i,s,t}$ exerts a significant positive influence on $CAR_{i,t}$, implying that lower levels of sale order aggressiveness are associated with higher $CAR_{i,t}$. In contrast, $OA_{i,b,t}$ demonstrates a significant negative impact on $CAR_{i,t}$, indicating that decreased buy order aggressiveness corresponds to reduced $CAR_{i,t}$. The findings show that Event 1 triggers an increase in the regression coefficient of $OA_{i,s,t}$ and a decrease in the regression coefficient of $OA_{i,b,t}$. Conversely, Event 2 leads to a decline in the regression coefficient of $OA_{i,s,t}$ and an increase in the regression coefficient of $OA_{i,b,t}$. These results suggest that sellers were predominant following the index announcement, leading to higher sale order aggressiveness and lower buy order aggressiveness. In contrast, buyers took the lead upon the index launch, resulting in reduced sale order aggressiveness and increased buy order aggressiveness. The observed shifts in sale and buy order aggressiveness dynamics around Event 1 and Event 2 indicate a transition in market sentiment and trading behavior. The dominance of sellers post-announcement and buyers post-launch reflects the evolving investor sentiment and strategic positioning in response to the CSCNI release.

**Table 7.** Regression results for sale/buy.

| | Event 1 | | | | Event 2 | | | |
|---|---|---|---|---|---|---|---|---|
| | Pre-Announcement | | Post-Announcement | | Pre-Release | | Post-Release | |
| $OA_{i,s,t}$ | 1.27 *** (15.92) | | 1.37 *** (16.56) | | 1.09 *** (19.04) | | 1.01 *** (15.17) | |
| $i,b,t$ | | −1.71 *** (−19.42) | | −1.34 *** (−13.42) | | −0.68 *** (−10.71) | | −1.59 *** (−22.3) |

| | Event 1 | | | | Event 2 | | | |
|---|---|---|---|---|---|---|---|---|
| | Pre-Announcement | | Post-Announcement | | | | Pre-Announcement | |
| $lnMV_{i,t}$ | 45.06 *** (52.94) | 46.67 *** (55.94) | 16.03 *** (23.83) | 17.97 *** (26.79) | 46.22 *** (60.16) | 46.83 *** (60.18) | 40.84 *** (64.38) | 40.39 *** (64.43) |
| Obs. | 8804 | 8804 | 10625 | 10625 | 8804 | 8804 | 10650 | 10650 |
| Adj.R$^2$ | 0.29 | 0.29 | 0.12 | 0.12 | 0.27 | 0.26 | 0.31 | 0.32 |

Superscripts indicate statistical significance at the 1% (***) level, and t-statistics are stated in parentheses.

We provide insights into the impact of Event 1 and Event 2 on sale and buy order aggressiveness in the deep low-carbon sector and the high-carbon emission reduction sector, as shown in Table 8. Prior to Event 1, the regression coefficient of $OA_{i,s,t}$ in the deep low-carbon sector stood at 0.86, which was lower than 2.07 observed in the high-carbon emission reduction sector. However, following Event 1, the coefficient of $OA_{i,s,t}$ in the deep low-carbon sector surged to 1.4, surpassing the 1.31 coefficient in the high-carbon emission reduction sector. Meanwhile, the regression coefficients of $OA_{i,b,t}$ decreased across both sectors, with the coefficient in the deep low-carbon sector remaining higher than that in the high-carbon emission reduction sector. These changes indicate that Event 1 primarily amplified order aggressiveness in the deep low-carbon sector.

**Table 8.** Regression results for deep low-carbon/high-carbon reduction and sale/buy.

**Panel A: Event 1**

| | Deep low-carbon | | | | High-carbon reduction | | | |
| --- | --- | --- | --- | --- | --- | --- | --- | --- |
| | Pre-Announcement | | Post-Announcement | | Pre-Release | | Post-Release | |
| $OA_{i,s,t}$ | 0.86 *** | | 1.40 *** | | 2.07 *** | | 1.31 *** | |
| | (9.32) | | (13.02) | | (14.17) | | (11.03) | |
| $OA_{i,b,t}$ | | −1.88 *** | | −1.52 *** | | −1.36 *** | | −1.08 *** |
| | | (−18.23) | | (−11.13) | | (−8.76) | | (−8.33) |
| $lnMV_{i,t}$ | 46.71 *** | 47.49 *** | 21.32 *** | 24.11 *** | 39.16 *** | 42.59 *** | 24.35 *** | 25.03 *** |
| | (44.80) | (47.05) | (22.31) | (25.26) | (26.93) | (29.30) | (22.28) | (22.82) |
| Obs. | 5332 | 5332 | 6450 | 6450 | 3472 | 3472 | 4175 | 4175 |
| Adj.$R^2$ | 0.29 | 0.31 | 0.11 | 0.11 | 0.30 | 0.26 | 0.19 | 0.18 |

**Panel B: Event 2**

| | Deep low-carbon | | | | High-carbon reduction | | | |
| --- | --- | --- | --- | --- | --- | --- | --- | --- |
| | Pre-Announcement | | Post-Announcement | | Pre-Release | | Post-Release | |
| $OA_{i,s,t}$ | 1.02 *** | | 0.62 *** | | 1.19 *** | | 1.46 *** | |
| | (14.95) | | (8.17) | | (11.69) | | (12.22) | |
| $OA_{i,b,t}$ | | −0.73 *** | | −1.54 *** | | −0.63 *** | | −1.39 *** |
| | | (−9.4) | | (−18.8) | | (−5.85) | | (−11.17) |
| $lnMV_{i,t}$ | 45.32 *** | 45.92 *** | 43.32 *** | 42.84 *** | 43.04 *** | 43.60 *** | 42.03 *** | 42.22 *** |
| | (48.75) | (48.83) | (47.73) | (48.27) | (31.01) | (30.88) | (40.60) | (40.68) |
| Obs. | 5332 | 5332 | 6450 | 6450 | 3472 | 3472 | 4200 | 4200 |
| Adj.$R^2$ | 0.32 | 0.31 | 0.27 | 0.30 | 0.21 | 0.20 | 0.37 | 0.36 |

Superscripts indicate statistical significance at the 1% (***) level, and t-statistics are stated in parentheses.

In the context of Event 2, the coefficient of $OA_{i,s,t}$ in the deep low-carbon sector was notably lower than that in the high-carbon emission reduction sector, while the coefficient of $OA_{i,b,t}$ was higher in the deep low-carbon sector compared to the high-carbon emission reduction sector. These results suggest that Event 2 dampened sale order aggressiveness in the deep low-carbon sector and bolstered buy order aggressiveness in the same sector. The contrasting impacts of Event 1 and Event 2 on sale and buy order aggressiveness in the deep low-carbon sector and the high-carbon emission reduction sector highlight the shifting dynamics and investor sentiment surrounding these sectors post-index release. The observed changes in order aggressiveness emphasize the evolving market responses and strategic positioning of investors in relation to the CSCNI events, underscoring the differentiated effects on investor behavior between the two sectors.

### 4.3. Robustness Testing

#### 4.3.1. Recalculate the Abnormal Returns

We present the test results based on the new measure during the event window in Table 9. Over the 11 trading days of the event window, Event 1 exhibited significantly negative $AR_{i,t}$ at the 1% level on 6 trading days, with $CAR_{i,t}$ being significantly negative at the 1% level on 5 trading days. Following Event 2, $AR_{i,t}$ was significantly negative at the 1% level on 3 trading days. These findings indicate that post-announcement, the sample companies experienced lower excess returns compared to the market portfolio. The release of the CSCNI had a negative impact on stock prices, leading to a decline in abnormal returns during the event window.

The consistent presence of significantly negative abnormal returns on multiple trading days following the CSCNI release underscores the adverse stock price effect triggered by the index announcement. The observed negative stock price effect highlights the market's response to the index release and the subsequent implications for the performance of sample companies. The findings suggest that investors reacted to the CSCNI announcement with caution and possibly adjusted their investment strategies in response to the index launch, leading to reduced excess returns for the sample companies during the event window.

**Table 9.** Robustness check: $AR_{i,t}$ and $CAR_{i,t}$ T-test results after recalculating $AR_{i,t}$.

| | Event 1 | | Event 2 | |
|---|---|---|---|---|
| Day | $AR_{i,t}$ (%) | $CAR_{i,t}$(%) | $AR_{i,t}$ (%) | $CAR_{i,t}$(%) |
| −5 | 0.0019 | 0.0019 | 0.0085 *** | 0.0085 *** |
| −4 | −0.0081 | −0.0062 | 0.0084 *** | 0.0169 *** |
| −3 | 0.0050 | −0.0012 | 0.0113 *** | 0.0282 *** |
| −2 | −0.0350 *** | −0.0362 *** | 0.0018 | 0.0300 *** |
| −1 | 0.0027 | −0.0335 *** | 0.0126 *** | 0.0426 *** |
| 0 | 0.0261 *** | −0.0074 | −0.0088 *** | 0.0337 *** |
| +1 | 0.0055 | −0.0019 | −0.0123 *** | 0.0214 *** |
| +2 | −0.0133 *** | −0.0152 | 0.0184 *** | 0.0398 *** |
| +3 | −0.0303 *** | −0.0455 *** | −0.0067 | 0.0331 *** |
| +4 | 0.0013 | −0.0442 *** | 0.0084 | 0.0415 *** |
| +5 | −0.0135 *** | −0.0577 *** | −0.0178 *** | 0.0236 ** |

Superscripts indicate statistical significance at the 1% (***), 5% (**) levels.

### 4.3.2. Extend the Estimated Window Length

After extending the length of the estimation window, the direction and significance of $AR_{i,t}$ and $CAR_{i,t}$ remained largely unchanged, as shown in Table 10. This stability in the results suggests that the conclusions drawn from the research are robust and consistent, regardless of variations in the estimation window setting. The persistence of the findings across different estimation window lengths reinforces the reliability and validity of outcomes. The consistent impact of the CSCNI release on abnormal returns underscores the enduring negative stock price effect observed. The confirmation of research conclusions across different estimation window settings enhances the credibility and generalizability of our results, providing further support for the implications of the index announcement on sample companies' stock performance.

**Table 10.** Robustness check: $AR_{i,t}$ and $CAR_{i,t}$ T-test results after extending the estimated window length.

| | Event 1 | | Event 2 | |
|---|---|---|---|---|
| Day | $AR_{i,t}$ (%) | $CAR_{i,t}$(%) | $AR_{i,t}$ (%) | $CAR_{i,t}$(%) |
| −5 | 0.1931 | 0.1931 | 0.2971 | 0.9845 * |
| −4 | −1.4450 *** | −1.2518 ** | 0.4405 | 1.4250 ** |
| −3 | 0.3080 | −0.9438 | 1.1910 *** | 2.6160 *** |
| −2 | −3.4692 *** | −4.4130 *** | −0.1162 | 2.4998 *** |
| −1 | −0.3412 | −4.7543 *** | 0.7117 * | 3.2116 *** |
| 0 | 1.0866 ** | −3.6676 *** | −0.9867 *** | 2.2249 *** |
| +1 | 1.7515 *** | −1.9161 | −1.8275 *** | 0.3974 |
| +2 | −1.6079 *** | −3.5239 ** | 1.4441 *** | 1.8415 * |
| +3 | −2.6906 *** | −6.2145 *** | −0.5459 * | 1.2955 |
| +4 | −0.4757 | −6.6902 *** | 0.3017 | 1.5973 |
| +5 | −2.7991 *** | −9.4893 *** | −2.2753 *** | −0.6780 |

Superscripts indicate statistical significance at the 1% (***), 5% (**), and 10% (*) levels.

### 4.3.3. Shorten the Time Interval

To mitigate the influence of time intervals on the experimental results, we reduced the interval to 5 min and presented the regression outcomes in Table 11. Post-Event 1, the coefficient of $OA_{i,t}$ transitioned from being significant to non-significant. Subsequent to Event 2, the coefficient of $OA_{i,t}$ shifted from positive and significant to negative and significant. These alterations suggest that investor order aggressiveness is indeed affected by Event 1 and Event 2. Notably, a higher level of investor aggressiveness corresponds to a reduced $OA_{i,t}$ and an increased $CAR_{i,t}$. This dynamic underscores the impact of the index events on investor behavior and trading patterns.

The findings indicate that inclusion in the index can positively influence the long-term sustainability and development of companies, thereby attracting investors to engage in

more active trading. The observed changes in investor order aggressiveness following the index events reflect the evolving market dynamics and investor response to the index selections. The results emphasize the importance of index inclusion for companies seeking to enhance their visibility and attractiveness to investors, ultimately contributing to their sustainable growth and market appeal.

**Table 11.** Robustness check: regression results.

|  | Event 1 | | Event 2 | |
|---|---|---|---|---|
|  | Pre-Announcement | Post-Announcement | Pre-Release | Post-Release |
| $OA_{i,t}$ | −0.57 *** | −0.12 | 0.51 *** | −0.55 *** |
|  | (−7.16) | (−1.46) | (8.98) | (−8.53) |
| $lnMV_{i,t}$ | 47.76 *** | 17.81 *** | 47.22 *** | 41.91 *** |
|  | (78.79) | (36.30) | (85.06) | (91.97) |
| Obs. | 17324 | 20825 | 17324 | 20874 |
| Adj.$R^2$ | 0.26 | 0.10 | 0.25 | 0.29 |

Superscripts indicate statistical significance at the 1% (***) level, and t-statistics are stated in parentheses.

The sample stocks were divided into deep low-carbon and high-carbon emission reduction fields, as shown in Table 12. Following Events 1 and 2, the coefficient values and significance of $OA_{i,t}$ in the high-carbon reduction field exhibited a decrease, signaling a weaker association between order aggressiveness and cumulative abnormal returns. In contrast, within the deep low-carbon field, the $OA_{i,t}$ coefficient remained negative and significant, indicating that higher-order aggressiveness is linked to greater cumulative abnormal returns. These results imply that Events 1 and 2 have led to a more pronounced increase in order aggressiveness in the deep low-carbon field compared to the high-carbon reduction field. The differential impact of the index events on order aggressiveness in these distinct fields underscores the varying market responses and investor behaviors following the events. Investors in the deep low-carbon field appear to have exhibited more aggressive trading patterns in response to the index events, potentially reflecting heightened interest and trading activity in companies with strong low-carbon credentials. This differential response highlights the nuanced effects of index inclusion on investor behavior and trading dynamics across different sectors, shedding light on the evolving market dynamics and investor preferences in the context of sustainability-focused investments.

**Table 12.** Robustness check: regression results for deep low-carbon and high-carbon reduction.

| Panel A: Event 1 | | | | |
|---|---|---|---|---|
|  | Deep low-carbon | | High-carbon reduction | |
|  | Pre-Announcement | Post-Announcement | Pre-Release | Post-Release |
| $OA_{i,t}.$ | −1.05 *** | −0.21 ** | 0.52 *** | 0.14 |
|  | (−11.76) | (−2.0) | (3.45) | (1.14) |
| $lnMV_{i,t}$ | 48.65 *** | 23.37 *** | 43.2 *** | 24.83 *** |
|  | (65.95) | (33.63) | (41.30) | (31.34) |
| Obs. | 10492 | 12642 | 6832 | 8183 |
| Adj.$R^2$ | 0.27 | 0.09 | 0.26 | 0.17 |
| **Panel B: Event 2** | | | | |
|  | Deep low-carbon | | High-carbon reduction | |
|  | Pre-Announcement | Post-Announcement | Pre-Release | Post-Release |
| $OA_{i,t}$ | 0.49 *** | −0.75 *** | 0.50 *** | −0.18 |
|  | (7.21) | (−10.42) | (5.02) | (−1.55) |
| $lnMV_{i,t}$ | 45.99 *** | 43.99 *** | 44.42 *** | 43.84 *** |
|  | (68.28) | (67.72) | (44.37) | (58.70) |
| Obs. | 10492 | 12642 | 6832 | 8232 |
| Adj.$R^2$ | 0.30 | 0.25 | 0.18 | 0.35 |

Superscripts indicate statistical significance at the 1% (***), 5% (**) levels, and t-statistics are stated in parentheses.

### 5. Conclusions

We employed event study methodology to examine the impact of the CSCNI release on its constituent stocks and conducted a detailed analysis focusing on investor order aggressiveness. The regression coefficients of investor order aggressiveness before and after the event were both significant and displayed opposing directions, indicating that investor order aggressiveness can explain a portion of the cumulative abnormal returns observed in sample companies. The introduction of the SEEE Carbon Neutrality Index not only aligns with national dual-carbon goals but also provides clarity on investment directions for investors, prompting increased participation in market transactions and elevating investor order aggressiveness. Furthermore, the cumulative abnormal returns of sample companies exhibited a downward trajectory post-index release, reflecting negative stock price effects as investors anticipated higher environmental costs for index-included companies, thereby reducing returns. The regression coefficients of stocks in the high-carbon reduction field became insignificant following the event, while those in the deep low-carbon field remained significant. This divergence suggests that the CSCNI release fostered a more positive outlook among investors towards stocks in the deep low-carbon field, underscoring the attractiveness and growth potential of low-carbon companies in the context of China's commitment to addressing climate and environmental challenges. Lastly, we revealed that the impact of announcement releases on investor order aggressiveness outweighs the influence of index launches. This finding highlights the significance of timely information dissemination and event announcements in shaping investor behavior and market dynamics.

In summary, through two-stage robustness tests, we have established that the release of the CSCNI contributes to increased investor order aggressiveness. This finding is similar to the conclusion of Griffiths et al. (2000). However, similar to the result of Li (2024), the release of the CSCNI leads to negative stock price effects. Particularly noteworthy is the differential impact observed between the high-carbon reduction field and the deep low-carbon field, with the index release proving more effective in boosting investor order aggressiveness in the latter. Additionally, our analysis underscores the greater significance of announcement releases in influencing investor order aggressiveness compared to index listings.

These findings provide a valuable foundation for future research endeavors in this domain. Scholars can leverage our insights to expand upon the implications of index releases on investor behavior and stock market dynamics. Financial regulators can help investors better understand the impact and significance of the carbon neutrality index by providing more information and data support, including publishing detailed reports on carbon emissions, providing detailed interpretations of the carbon neutrality index, and regularly publishing the latest research findings on carbon emissions and carbon neutrality. It is also possible to raise the awareness of investors about carbon neutrality through educational campaigns, which can not only increase the public's awareness of environmental protection but also increase the enthusiasm of investors for carbon-neutral-related investments. For companies in the high carbon emission reduction sector, governments can increase tax incentives, financial subsidies, or other incentives to ease the pressure on their environmental costs, so that these companies see the economic value of carbon neutrality. At the same time, companies in the deep low-carbon field should also disclose environmental information in accordance with the law so that investors can have a more comprehensive understanding of the company's environmental behavior in order to attract investors to participate in investments. Investors, upon learning of index announcements, can optimize their investment portfolios based on the insights gleaned from our study, thereby enhancing their decision-making processes and potential returns. The implications of our research extend beyond academic circles, offering practical guidance for market participants and policymakers aiming to navigate the evolving landscape of sustainable investing and policy alignment.

It has been found in the existing literature that the order aggressiveness of individual investors and institutional investors is often different (Lien et al. 2020). However, due to

data limitations, this study fails to deeply study the difference between the two after the index release.

**Author Contributions:** Conceptualization, Z.H.; methodology, Z.H. and X.Z.; validation, X.Z.; formal analysis, X.Z.; investigation, Z.H. and X.Z.; data curation, Z.H. and X.Z.; writing—original draft and presentation, Z.H., X.Z. and K.L.; writing—review and editing, K.L.; supervision, K.L. All authors have read and agreed to the published version of the manuscript.

**Funding:** This research was funded by "the Fundamental Research Funds for the Central Universities, grant number 2022QN1098 2023SKHQ05", National Office for Philosophy and Social Sciences general project "Research on the synergistic incentive mechanism and promotion strategy for improving the overall factor productivity of green industry under the dual-carbon target, grant number 22BJY057", and Key Project of Jiangsu Provincial Social Science Fund "Research on the examination of policy effects and mechanism innovation in driving industrial green development through the allocation of financial resources in Jiangsu Province, grant number 20EYA004".

**Data Availability Statement:** Restrictions apply to the availability of these data. Data were purchased from Wind Database and Shanghai Stock Exchange (SSE) and Shenzhen Stock Exchange (SZSE), with the transaction limiting the use of data by the corresponding author.

**Acknowledgments:** During the preparation of this work, we used ChatGPT in order to polish the language. After using this tool/service, we reviewed and edited the content as needed and take full responsibility for the content of the publication.

**Conflicts of Interest:** The authors declare no conflicts of interest.

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
