# Peer review of "The Impact of CSI SEEE Carbon Neutral Index Launched on Order Aggressiveness"

_jrfm, doi:10.3390/jrfm17050198_

Round 1

Reviewer 1 Report

Comments and Suggestions for Authors

I really appreciate the authors’ contribution to studying the impact of the CSI SEEE Carbon Neutral Index (CSCNI) launch on order aggressiveness.

1. The paper is generally well written with adequate clarity which demonstrates the author’s knowledge in the field.

2. The paper structure respects the structure requested, in general, in research papers.

3. The abstract presents the aim of the paper, the analysis method and the main results (conclusion) described in the manuscript.

4. The Introduction section presents the context (scope of the paper), similar studies, aim, main results and paper structure.

I recommend highlighting some research questions, but more importantly the novelty of the paper. An important limitation of the paper, in its current form, is that is very difficult know the original aspects of the paper. This has to be clear from the beginning, in introduction. In a trivial form, the message is there somehow, but it not so straightforward, you must read it between the lines. And this added value should go beyond the geographical scope of the study, in order to be meaningful.

The authors mention: “we introduce the concept of investor order aggressiveness” (lines 82-83). But they do not introduce the concept, they merely use it. It would help to have here at least a significant reference who firstly introduced the concept. Then extensively explain the concept in H1 (see next comment).

The authors make general claims without specifying who made those claims. For example, “there is anticipation that China's green financial product system will be further strengthened.” (lines 79-80). Who anticipated? Where is this information extracted from? If the authors are the ones who anticipate, then they must rephrase: “We strongly believe that..”. Or, “Investors are concerned that inclusion in the index may lead…” Who are we talking about? Who are these investors specifically? Or, who declared that? Or, who researched on that? General claims, if there are not ‘common locus’, must be backed up by sources.

5. The literature review from the Hypothesis section could use some refinement.

For example, H1, which I believe is the most important hypothesis from this research, is built on solely the work of Kim et al. (2014) [so, from a decade ago] who conducted “a study on social responsibility disclosure” [a topic which only touches the green discourse]. Here, the authors should extensively discuss about the “investor order aggressiveness” concept: who introduced it (?), who used it (?), what kind of results other researchers obtained (?), why is this better then “the traditional proxy variables like the Baidu Index and Google search index” (?) and in what regards specifically? I strongly believe that this hypothesis, in particular, needs much more attention. And this is even more relevant, when considering that the concept is included in the title.

The same goes for H3, which is built upon only one work: Lu et al. (2024).

6. The methodology is well presented and has a dedicated section.

The authors should provide some details with regard to what Ask1, Ask5, Ask10 mean.

7. The results are clearly presented, no doubt about it (from an econometrical perspective). Nonetheless, I consider that a new and dedicated section (Discussions) would be useful where the socio-economic interpretation of the results would be extensively tackled. The Results section includes such comments, but the study gives the idea that the ‘story’ is missing. And the ‘story’ is the one that sells. I understand the model but I believe that it lacks the story behind it. In what way these results could be used practically (pushing thus the theory forward and the level of understanding). What are the implications, from an investor’s point of view, of confirming/rejecting the hypotheses? This dedicated section would be the most valuable part of the paper, one that would lay the groundwork for the added value of the paper.

8. I strongly recommend that the authors highlight better the implications or applicability in the Conclusion section. It’s true that they mention some contributing effects, but I believe that this could be done much better. Concrete, sound, clear recommendations with reference to existing policies or policies that are current on public debate. What exactly are the changes they expect in terms of policy measures? The final paragraph is somehow useful, but at the same time quite vague.

9. References are appropriate with the paper’s aim and scope.

10. I recommend that the authors present the limitations of the paper (in the conclusion section).

All in one, I really enjoyed reading the paper. I consider it to have a lot of potential. It encompasses good and relevant literature, a good model, the math is ok, but it lacks the story to bring it all together.  

Comments on the Quality of English Language

The text is fine. Some minor adjustments have to be made. For example, “Inclusion in the index signifies’’, inclusion of what? Of course, we know the answer, but this should not be deduced, but clearly emphasized. But, overall, only small adjustments are needed.

Reviewer 2 Report

Comments and Suggestions for Authors

1. The contribution of this study should be clearly highlighted in the abstract and introduction section. What contribution this study will make in the field of green finance indices. How the findings of this study will help and improve the understanding of these green finance indices?

2.  No specific policy recommendations are made based on the findings of this study.

3.   Motivation of conducting this study should be specified.

4.  Sample firms are not clearly discussed in the abstract.

5.  If authors refer/support the study with some theory such as “Efficient market hypothesis” or any behavioral theory, it will add more value to their work. In the current form, the paper fails to present any depth and context. So, theoretical justification is weak. I recommend authors to either refer to efficient market hypothesis or behavioral finance theories to give it to some context and to provide the study some strong theoretical foundations.

6.   In addition to the above mentioned theoretical justifications, authors should also discuss the ESG framework and discuss how investor responds to these environmental considerations. 

7.   Need further discussion on sample selection. How do authors justify that sample bias is not a problem in their study?

8.  Need justification on the following point

·       The justification for choosing 30 days’ estimation window? (any source/reference)

·       Which model was used for event study? Market model, mean adjusted, factor etc

·       Why no controls such as size, profit or market conditions are not taken while applying regression. To control for these factors may increase the significance of these results.

9.  A through discussion on the results is lacking in the article. Findings should be discussed and properly explained in the context of objective of this research. Moreover, the implications should also be highlighted and comparison of your findings with the previous studies should be made.   

Comments on the Quality of English Language

Acceptable but can be improved 

Round 2

Reviewer 1 Report

Comments and Suggestions for Authors

With regard to the paper at hand, I appreciate that the authors considered my suggestions. Only the novelty of the paper should have received more attention in the Introduction section. This could be deduced from the Abstract, but could have been as well provided in the Introduction.